🔓 | **Open Peer Review** | Clinical Microbiology | Methods and Protocols

# An integrated molecular approach for the detection and species-level identification of *Leishmania* spp. in clinical samples

Marikena Guadalupe Risso,[1] Wendy Lorena Quintero-Garcia,[1] Juan Jose Lauthier,[1] Giuliana Saggion,[1] Carlos Parra,[1] Catalina Gauder,[2] German Astudillo,[2] Sofia Echazarreta,[2] Monica Beatriz Foccoli,[3] Ana Andrea Pisarevsky,[3] Silvia Analia Repetto,[3,4] Paula Ruybal[1]

**ABSTRACT** Leishmaniases are parasitic diseases transmitted by infected sand flies, present in over 98 countries. Accurate diagnostic tools are critical for prevention and control, especially in non-endemic areas. We present a standardized molecular workflow combining two PCR-based strategies: sensitive detection through amplification of conserved *Leishmania* kinetoplast DNA (kDNA) minicircles and species identification via sequencing of a 1,116-bp *hsp*70 gene fragment. The optimized kDNA-PCR achieved detection limits of 0.05 parasites/mL for *Leishmania (Leishmania) infantum* and 0.5 parasites/mL for *Leishmania (Viannia) braziliensis*, the most prevalent species in Argentina. The method also enabled differential diagnosis with Chagas disease and endemic mycoses. Evaluation with 150 clinical samples from 83 patients showed 69.7% sensitivity and 68% specificity, reflecting improved performance compared to microscopy. Eight kDNA-PCR "false positives" were confirmed as true infections by *hsp*70 sequencing. Four species were identified: *Leishmania (Viannia) braziliensis*, *Leishmania (Viannia) guyanensis*, *Leishmania (Viannia) panamensis*, and *Leishmania (Leishmania) mexicana*. This workflow provides a reliable diagnostic and species-level tool in resource-limited, endemic settings.

**IMPORTANCE** Leishmaniases are neglected tropical diseases that remain a major public health problem in many parts of the world. Accurate diagnosis is essential for guiding treatment and controlling transmission, yet current methods often fail to detect low parasite loads or identify the infecting species. This study introduces a standardized molecular workflow that combines two complementary PCR-based strategies for the detection and species identification of *Leishmania* directly from clinical samples. The approach increases sensitivity, allows discrimination from other endemic infections, and identifies the circulating species with high reliability. Its implementation revealed multiple *Leishmania* species in Argentina and neighboring countries, highlighting its value for epidemiological surveillance. The workflow is simple, adaptable, and cost-effective, making it suitable for laboratories in resource-limited settings. By improving diagnostic accuracy, this method supports better patient management and strengthens regional capacity to monitor and control leishmaniasis.

**KEYWORDS** *Leishmania* spp., leishmaniasis, molecular diagnosis, species typing, neglected tropical diseases, parasitology, Argentina

Leishmaniases are vector-borne parasitic diseases caused by protozoan parasites of the order Kinetoplastida and family Trypanosomatidae. They constitute a significant global health concern and are among the most prevalent neglected tropical diseases, with over 1 billion people at risk of infection in more than 98 endemic countries. Between 700,000 and 1 million new cases are reported annually worldwide (1).

**Peer Reviewers** Estefania Calvo Alvarez, Universita degli Studi di Milano, Milan, Italy; Claudio Vieira da Silva, Universidade Federal de Uberlandia, Uberlandia, Brazil

Address correspondence to Paula Ruybal, paula.ruybal@hospitalitaliano.org.ar.

Marikena Guadalupe Risso and Wendy Lorena Quintero-Garcia contributed equally to this article. The order of these authors was determined based on their overall contribution to the study, with the first author contributing more substantially to the original study design.

The authors declare no conflict of interest.

See the funding table on p. 16.

A recent review by Akhoundi et al. (2) raised taxonomic concerns by describing 54 *Leishmania* species distributed across four subgenera: *Leishmania*, *Viannia*, *Sauroleishmania*, and *Mundinia*. Of these, 31 are known to infect mammals, and at least 21 are pathogenic to humans (3). This wide diversity of species contributes to clinical manifestation variability of leishmaniasis, which includes cutaneous (CL), mucocutaneous (MCL), and visceral (VL) forms. In the Americas, the cutaneous and mucocutaneous manifestations are identified as American Tegumentary Leishmaniasis (ATL) (4).

CL typically presents as single or multiple skin lesions, starting as macules that evolve into papules and eventually ulcers, with regular and raised borders. MCL involves the mucous membrane of the upper respiratory and/or digestive tracts, typically emerging months or years after CL. Finally, VL, the most severe form, affects hematopoietic organs and is characterized by fever, splenomegaly, and hepatomegaly, with severe systemic symptoms (5).

According to the World Health Organization (WHO), 71 out of 200 countries and territories (36%) are endemic for both CL and VL. Human cases have been reported across the Americas, from the southern United States to northern Argentina, with the exception of the Caribbean islands and Chile. Between 2013 and 2023, a total of 482,611 CL cases and 2,039 imported CL cases were reported to the WHO (6). In 2023, 91.6% of reported cases were CL, and 5.4% were MCL, the highest numbers since 2012, with increasing trends in the Americas, particularly in Paraguay, Bolivia, and Peru. The geographic expansion of VL within and across countries was also observed in 2023, especially along the borders of Argentina, Bolivia, Brazil, and Paraguay. Brazil bears the highest disease burden in the region (91%). However, its proportional contribution has been declining over the years, while Argentina, Paraguay, and Venezuela have shown an upward incidence trend (7).

In Argentina, leishmaniases have become an increasing public health challenge in recent decades. ATL, endemic in 10 provinces, has experienced a significant increase, while the first autochthonous cases of VL were reported in 2006 in the city of Posadas (Province of Misiones). Since then, the disease has spread from autochthonous transmission foci (8). According to PAHO, there are six countries with geographic VL expansion: Argentina, Bolivia, Brazil, Guatemala, Honduras, and Uruguay (5).

To date, four *Leishmania* species have been identified as the causative agents of ATL in Argentina: *Leishmania (Viannia) braziliensis*, *Leishmania (Leishmania) amazonensis*, *Leishmania (Viannia) guyanensis*, and *Leishmania (Viannia) panamensis*, with the first being the most prevalent species. Furthermore, *Leishmania (Leishmania) infantum* has been associated with cases of VL and canine leishmaniasis (8–12).

The diversity of species and clinical manifestations, combined with the challenge of obtaining optimal clinical samples, makes diagnosis complex. Direct methods allow confirmation of any suspected or probable leishmaniasis case if one or more of the following techniques yield a positive result: direct microscopic search for amastigotes (tissue from cutaneous and/or mucocutaneous lesions, or in biopsies of bone marrow, spleen, liver, or lymph nodes), *in vitro* culture of promastigotes from skin lesions or lymph node aspirates, or biopsies of skin or mucosal lesions, or PCR on tissue samples (13).

One of the most effective PCR targets is kinetoplast DNA (kDNA), which consists of two types of circular DNA molecules: minicircles (~1 kb, approximately 10,000 copies) and maxicircles (~23 kb, 25–50 copies). Its high copy number and sensitivity have made kDNA the most widely used amplification target for *Leishmania* detection.

However, although species identification is not a routine procedure in Argentina, it is crucial given the diversity of *Leishmania* parasites in the region, as the infecting species determines disease pathology and plays a fundamental role in guiding appropriate treatment and prognosis (14). It has been demonstrated that assays amplifying the conserved region of the kDNA minicircle or the small subunit of rDNA are the most sensitive tests but only identify *Leishmania* parasites at the genus and/or subgenus level (15). Markers with sensitivity levels suitable for their use in clinical samples and widely used for species determination are the genes encoding Heat Shock Protein 70

type I (hereafter referred to as *hsp*70), 3′-untranslated region of HSP70 (type I), and cytochrome b (*cyt*B) (16, 17). However, *hsp*70 is the most abundant and robust species marker available in public-access databases (18).

In summary, leishmaniases represent a persistent challenge to global health, with a wide species diversity leading to variability in clinical manifestations. In this context, we hypothesize that combining a diagnostic PCR targeting kinetoplast DNA with *hsp*70-based molecular typing constitutes a more sensitive and specific approach for both parasite detection and species identification, representing a useful tool for infection diagnosis and epidemiological surveillance. In this study, we evaluate a diagnostic algorithm that not only proposes a PCR for the molecular detection of the parasite but also includes the determination of the species involved in each clinical case, emphasizing its importance in therapeutic decision-making.

## RESULTS

### Standardization of kDNA-PCR

The amplification of the conserved region of the kDNA minicircle (kDNA-PCR) was optimized by addressing three key aspects. First, the new primer design considered both the *in silico* evaluation of reported primers against *Leishmania* species in the NCBI database and their high Tm difference (8°C), prompting sequence modification (kDNA22-FN and kDNA22-RN) (19). Second, we optimized the composition of the PCR reaction mixture (primers and $MgCl_2$ concentrations), cycling conditions (annealing temperature and number of cycles), and incorporated an internal reaction control (human *β-globin* gene) in a multiplex format, anticipating its use in clinical sample evaluation. Third, optimization was performed using DNA from reference strains of the most relevant *Leishmania* species in Argentina (Fig. 1). The amplification products were sequenced, and their identity was verified using BLASTn (data not shown).

### Specificity of kDNA-PCR

The kDNA-PCR demonstrated high specificity for *Leishmania* spp., with no cross-detection of pathogens genetically related and/or displaying similar clinical presentations in shared endemic regions (differential diagnosis) (Fig. 2).

The presence of DNA from those pathogens in the tested samples was confirmed using specific PCRs targeting a conserved 600 bp region of the fungal ribosomal ITS gene or the 188 bp TCZ nuclear minisatellite region of *T. cruzi* (data not shown). Moreover, DNA extracted from uninfected human blood rendered no amplification products, thus supporting the specificity of kDNA-PCR (Fig. 1).

### Analytical sensitivity of kDNA-PCR

To determine the limit of detection (LoD) of kDNA-PCR, the complete protocol was applied to standard parasitemia curves of *L. (L.) infantum* and *L. (V.) braziliensis*, the most prevalent *Leishmania* species in the country. The LoD for each species was defined according to positivity percentages for each point on the curves. Thus, the LoD achieved was 0.5 parasites/mL for *L. (V.) braziliensis* and 0.05 parasites/mL for *L. (L.) infantum*.

### Diagnostic performance of kDNA-PCR

The qualitative PCR diagnostic protocol was evaluated in 83 patients suspected of having CL, MCL, or VL, including eight follow-up cases (Fig. 3).

A total of 150 clinical samples (mainly smears, biopsies, and swabs) were studied, corresponding to 83 patients: 27 were female (32.53%, mean age = 48 years), 53 were male (63.86%, mean age = 50 years), and 3 had undetermined gender. Regarding the endemic areas involved, each patient informed their place of origin or previous visits/stays in leishmaniasis-endemic areas. This allowed us to estimate that 32.53% of the patient population likely acquired the infection in Argentina (*N* = 27), 10.84% in Bolivia (*N*

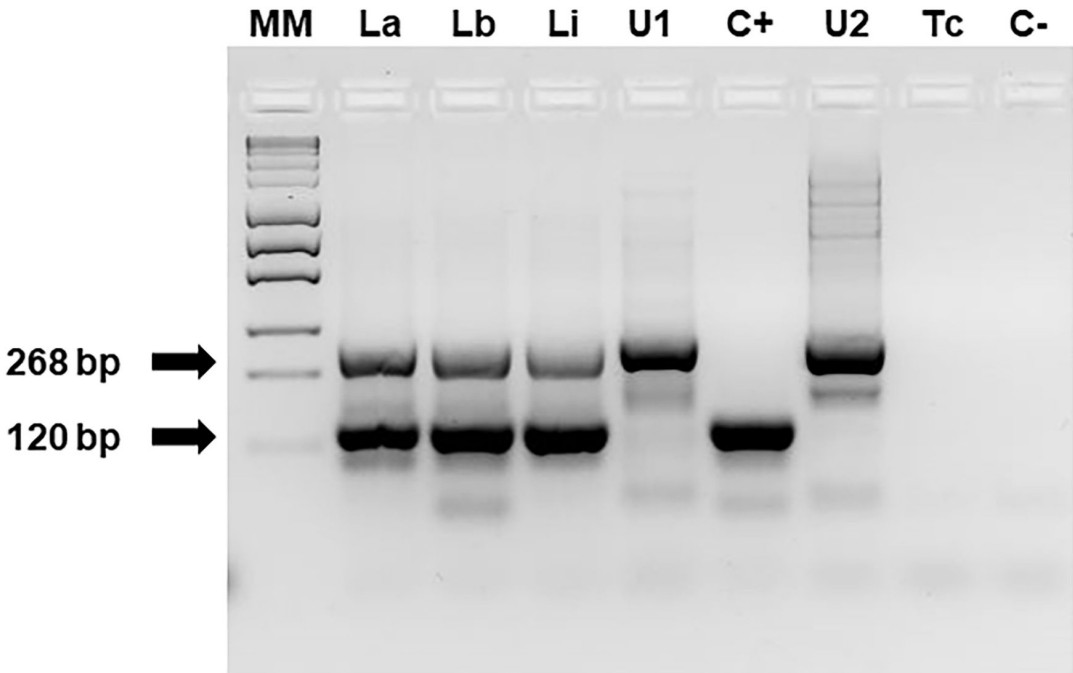

**FIG 1** Optimized kDNA-PCR assay. Reaction mixture according to section "Optimization of amplification conditions" (2.5 mM MgCl$_2$ and 0.5 µM each kDNA primer). La, *L. (L.) amazonensis*; Lb and C+, *L. (V.) braziliensis*; Li, *L. (L.) infantum*; U1 and U2, uninfected individuals' blood DNA; Tc: *Trypanosoma cruzi*; C–, no template DNA; and MM, MassRuler Express Forward DNA ladder Mix (ThermoFisher). PCR products from the *β-globin* gene (268 bp) and kDNA minicircle (120 bp) are indicated by black arrows.

= 9), 1.2% in Brazil (*N* = 1), 1.2% in Colombia (*N* = 1), 1.2% in Costa Rica (*N* = 1), 1.2% in Mexico (*N* = 1), 1.2% in Panama (*N* = 1), 14.46% in Paraguay (*N* = 12), and 2.41% in Peru (*N* = 2). Additionally, 9.64% of the patients (*N* = 8) reported multiple previous stays in any of the aforementioned countries except Mexico. This is mainly because they are seasonal workers, often employed in rural or construction activities, who have resided in several leishmaniasis-endemic regions. Among the Argentine cases, the reported provinces of origin or exposure included Misiones, Salta, Santiago del Estero, Formosa, Tucumán, Chaco, and Jujuy (see Table S1 at https://doi.org/10.5281/zenodo.18854787). Neverthe-less, information regarding the suspected place of infection could not be obtained for 24.10% of the patients (*N* = 20). Regarding clinical suspicion, 37.35% (*N* = 31) of the patients showed clinical manifestations compatible with MCL, and 48.19% (*N* = 40) with CL. Among the patient population, clinical suspicions of two of the less common forms of CL were also reported: multiple (15.66%, *N* = 13) and diffuse (2.41%, *N* = 2). Another patient (1.20%) reported symptoms and clinical signs compatible with both MCL and CL and was considered within both separate groups (MCL and CL) for kDNA-PCR test performance analyses, while 12.05% (*N* = 10) showed clinical manifestations compatible with VL. These values were estimated considering one patient for whom clinical suspi-cion data were unavailable but was still included (1.20%). Furthermore, eight patients (21 samples) were followed up for days/months after treatment with amphotericin B or meglumine antimoniate (see Table S1 at https://doi.org/10.5281/zenodo.18854787).

Figure 3 shows that all patients were diagnosed through the current reference method, resulting in a disease prevalence of 39.76% in the studied population.

The diagnostic performance of kDNA-PCR was evaluated using 150 samples from 83 patients, including follow-up samples; however, its well-known limitations of the reference method, including operator dependency and suboptimal sensitivity and specificity, affected kDNA-PCR performance estimation. PCR-positive but smear-negative cases may represent true infections missed by microscopy, resulting in an apparent decrease in PCR specificity. Such imperfections in the reference standard can introduce

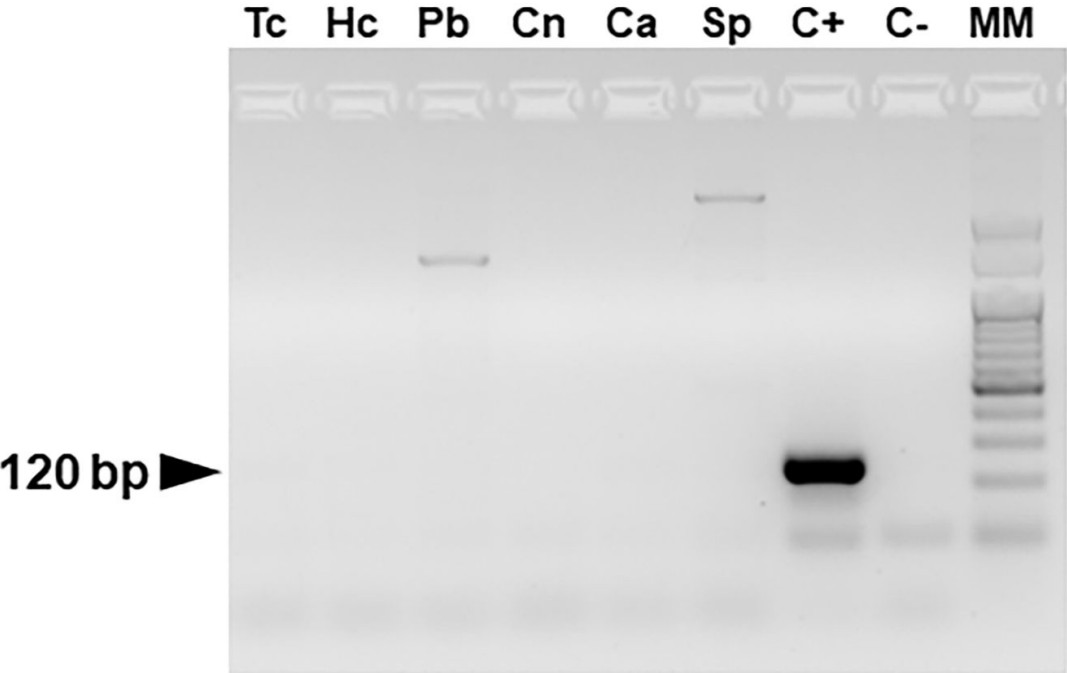

**FIG 2** Evaluation of kDNA-PCR specificity for *Leishmania* spp. detection. Tc, *Trypanosoma cruzi*; Hc, *Histoplasma capsulatum*; Pb, *Paracoccidioides brasiliensis*; Cn, *Candida nivariensis*; Ca, *Candida albicans*; Sp, *Sporothrix brasiliensis*; C+, *L. (V.) braziliensis*; C−, no template; and MM, 50 bp ladder, INBIO Highway. The amplification of kDNA-PCR (120 bp) is indicated by the black arrow.

misclassification bias, influencing sensitivity and predictive values. Notably, the low specificity of smear microscopy could have led to both an underestimation of PCR sensitivity, by misclassifying true negatives as false negatives, and an overestimation of PCR false negatives overall.

Based on the binomial distribution, a minimum of 83 individuals were required to estimate a 39.8% proportion with 95% confidence and a 4% margin of error. Wide confidence intervals reflect this sample size constraint.

Overall sensitivity and specificity were 69.70% and 68.00%, respectively (Table 1). In total, 16 patients tested positively only by PCR and 10 only by microscopy; notably, 2 of the latter were later confirmed to be infected by *Paracoccidioides* spp., possibly leading to misidentification under the microscope. Similar values were obtained when analyzing only samples collected at the time of diagnosis (*N* = 129). However, when considering the entire sample set (*N* = 150), overall specificity increased to 77%, driven by a 100% improvement in specificity in the follow-up samples, which showed a narrower confidence interval and reduced dispersion (Table 1).

To further the analysis, we evaluated the results from both the patient and sample populations according to the clinical suspicion recorded at diagnosis and follow-up, aiming to assess whether the diagnostic technique showed any performance bias related to the suspected clinical presentation of leishmaniasis (spectrum bias). However, statistical analysis could not be performed for samples suspected of VL, or for follow-up samples from CL and MCL cases, due to insufficient sample sizes (*N* = 12, *N* = 7, and *N* = 14, respectively). Diagnostic performance metrics showed similar ranges to those observed in the previous analysis, with a notably low specificity in the diagnosis of MCL (Table 2). This may be attributed to the low parasite load in these samples, where the molecular method is more sensitive than the reference method.

Moreover, the performance of the diagnostic test was evaluated according to the type of sample used. As shown in Fig. 3, kDNA-PCR was tested on DNA obtained from different sample settings (*N* = 150), such as swab (38%, *N* = 57), biopsy (26.66%, *N* = 40), smear (26%, *N* = 39), urine (0.67%, *N* = 1), bone marrow (BM) (5.33%, *N* = 8), cerebrospinal fluid

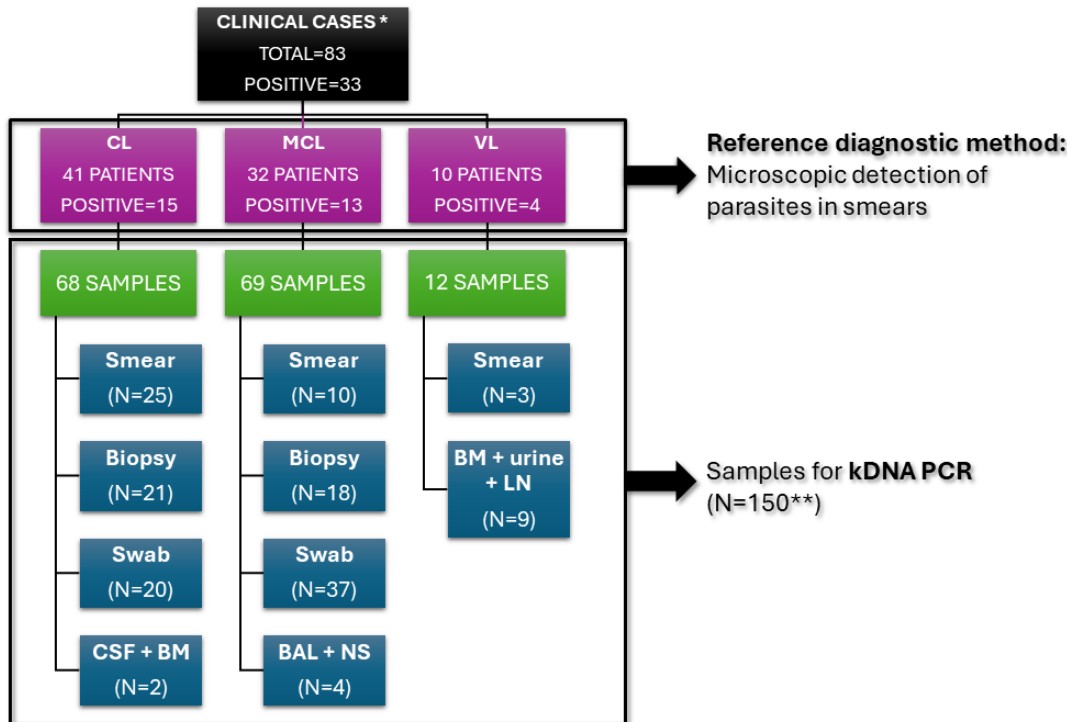

**FIG 3** Clinical cases suspected of cutaneous leishmaniasis (CL), mucocutaneous leishmaniasis (MCL), or visceral leishmaniasis (VL) and samples considered for the statistical analysis of the kDNA-PCR performance. BAL, bronchoalveolar lavage; NS, nasal secretion; CSF, cerebrospinal fluid; BM, bone marrow; and LN, lymph node. *: Includes one patient with no clinical suspicion data available and one patient with both CL and MCL lesions considered in both CL and MCL groups. **: Includes one smear from a patient with no clinical suspicion data available.

(CSF) (0.67%, $N = 1$), rhinorrhea (1.33%, $N = 2$), and bronchoalveolar lavage (BAL) (1.33%, $N = 2$). Specific statistical parameters were calculated only for biopsy, swab, or smear, the most frequent formats ($N = 136$). Follow-up and multiple samples per patient were included to increase the number of determinations (Table 3). Biopsy and swab settings demonstrated the highest specificity, while smear samples showed increased sensitivity. For smears, the same pattern observed in MCL diagnosis was noted: the limited sample volume, combined with a low parasite burden, makes the molecular method more efficient than the reference method in detecting the pathogen, thereby affecting the estimated specificity.

### Species typing of *Leishmania* spp.

All kDNA-PCR positive samples (54 for diagnostic evaluation and 6 for follow-up) were subjected to *hsp*70 sequence analysis for species determination. For this final stage of the diagnostic algorithm, the *hsp*70 type I gene, which is conserved across *Leishmania* species, was targeted using the primer set previously described, with modification of the reverse primer based on available *hsp*70 type I sequences to optimize amplification across diverse species (20). This locus was selected considering known differences in *hsp*70 genomic organization among *Leishmania* species, including the absence of the *hsp*70 type II gene in *L. (V.) braziliensis*, as previously described (21).

The species determination protocol was successfully applied to 35 samples (31 for diagnostic evaluation and 4 for follow-up) from 26 diagnosed patients (see Table S1 at https://doi.org/10.5281/zenodo.18854787). Among them, one patient was monitored after amphotericin B treatment and another after meglumine antimoniate treatment. PCR amplification products were sequenced and compared to identify distinct *hsp*70 sequence variants among patient-derived samples. A total of nine *hsp*70 variants were defined based on sequence differences, reflecting intra- and interspecific diversity

**TABLE 1** Statistical analysis of performance of kDNA-PCR based on individual patient results (*N* = 83) and the complete sample population, both as a whole (*N* = 150) and considering samples collected at the time of diagnosis (*N* = 129) or during post-treatment follow-up (*N* = 21)[a]

| Patient's population | | | | | | |
|---|---|---|---|---|---|---|
| | N | Sn (%) | Sp (%) | PPV (%) | NPV (%) | P (%) |
| Patient's diagnosis | 83 | 69.70 | 68.00 | 58.98 | 77.27 | 39.76 |
| | | (51.29–84.41) | (53.30–80.48) | (47.51–69.54) | (66.21–85.51) | |
| Total samples | | | | | | |
| Total samples | 150 | 70.00 | 77.00 | 66.76 | 79.54 | 33.33 |
| | | (55.39–82.14) | (67.51–84.83) | (57.34–75.02) | (71.53–85.75) | |
| Total samples vs diagnosis and follow up | | | | | | |
| Samples for diagnosis | 129 | 67.44 | 73.26 | 62.47 | 77.32 | 33.33 |
| | | (51.46–80.92) | (62.62–82.23) | (52.56–71.43) | (68.52–84.23) | |
| Samples for follow up | 21 | 85.71 | 100.00 | 100 | 91.38 | 33.33 |
| | | (42.13–99.64) | (76.84–100.00) | (54.07–100.00) | (63.34–98.49) | |

[a]Sn, sensitivity; Sp, specificity; PPV, positive predictive value; and NPV, negative predictive value. Confidence intervals (95% CI) are reported in parentheses. *P*, prevalence of each subgroup.

(see Table S1 at https://doi.org/10.5281/zenodo.18854787). Species assignment was subsequently established through phylogenetic analysis by clustering clinical sequences with 94 global reference strains (Fig. 4).

From the analysis of this marker, four *Leishmania* species were identified in the clinical samples processed in this study: *L. (V.) braziliensis* (*N* = 27), *L. (V.) guyanensis* (*N* = 1), *L. (V.) panamensis* (*N* = 5), and *L. (L.) mexicana* (*N* = 2). *L. (V.) braziliensis* was the most frequently detected species in clinical samples, consistent with its prevalence in Argentina. Other species were identified in patients with epidemiological backgrounds consistent with the detected species, as follows: sample M107 [*L. (L.) mexicana*] corresponded to a patient living in Tulum, Mexico; sample M110 [*L. (V.) guyanensis*] was from a patient who had traveled to the Bolivian jungle; and samples M111, M130, M133F, and M134F [*L. (V.) panamensis*] were from patients who had traveled to Panama or Costa Rica. Sample M128 [*L. (L.) mexicana*] was classified as an imported case of unknown origin. Additionally, sample M129F, also *L. (V.) panamensis*, corresponded to a patient from Salta Province, Argentina, where this species had previously been detected by PS-PCR for the first time in the country (11).

The most frequent variant was HSP70-24 (18 samples from 14 patients). It was widely distributed both within Argentina (Chaco, Jujuy, Salta, and Santiago del Estero provinces) and across the region (Bolivia, Paraguay, and Peru), associated with MCL, CL, and diffuse CL (DfCL) cases (see Table S1 at https://doi.org/10.5281/zenodo.18854787). Neither heterozygosity nor coinfection was detected in this variant, which was resolved as the most frequent haplotype (HP22). This variant was predominantly present in MCL

**TABLE 2** Statistical analysis of the optimized kDNA-PCR performance by type of clinical suspicion: mucocutaneous leishmaniasis or cutaneous leishmaniasis (CL)[a]

| Patient's population | | | | | | |
|---|---|---|---|---|---|---|
| | N | Sn (%) | Sp (%) | PPV (%) | NPV (%) | P (%) |
| Diagnosis of CL | 41 | 73.33 | 73.08 | 64.26 | 80.59 | 36.59 |
| | | (44.90–92.21) | (52.21–88.43) | (47.09–78.41) | (63.47–90.84) | |
| Diagnosis of MCL | 32 | 76.92 | 52.63 | 51.73 | 77.56 | 40.62 |
| | | (46.19–94.96) | (28.86–75.55) | (37.98–65.23) | (53.98–91.05) | |
| Total samples vs clinical form | | | | | | |
| Samples for CL diagnosis | 68 | 63.16 | 79.59 | 67.13 | 76.60 | 27.94 |
| | | (38.36–83.71) | (65.66–89.76) | (51.58–79.66) | (64.11–85.71) | |
| Samples for MCL diagnosis | 55 | 76.19 | 64.71 | 58.76 | 80.46 | 38.18 |
| | | (52.83–91.78) | (46.49–80.25) | (46.01–70.44) | (64.81–90.20) | |

[a]Sn, sensitivity; Sp, specificity; PPV, positive predictive value; and NPV, negative predictive value. Confidence intervals (95% CI) are reported in parentheses. *P*, prevalence of each subgroup.

**TABLE 3** Statistical analysis of kDNA-PCR performance by sample setting (swab, smear, or biopsy)[a]

| Sample settings | | | | | | |
|---|---|---|---|---|---|---|
| | N | Sn (%) | Sp (%) | PPV (%) | NPV (%) | P (%) |
| Biopsy | 40 | 78.57 | 88.46 | 81.80 | 86.22 | 35.00 |
| | | (49.20–95.34) | (69.85–97.55) | (59.96–93.10) | (69.44–94.51) | |
| Swab | 57 | 63.64 | 91.43 | 83.05 | 79.21 | 38.60 |
| | | (40.66–82.80) | (76.94–98.20) | (61.35–93.80) | (68.47–86.98) | |
| Smear | 39 | 81.82 | 50.00 | 51.92 | 80.64 | 28.21 |
| | | (48.22–97.72) | (30.65–69.35) | (40.46–63.19) | (52.99–93.90) | |
| Sample settings vs diagnosis | | | | | | |
| Biopsy | 35 | 75.00 | 86.96 | 79.15 | 84.05 | 34.29 |
| | | (42.81–94.51) | (66.41–97.22) | (55.70–91.97) | (66.13–93.43) | |
| Swab | 44 | 61.11 | 88.46 | 77.76 | 77.51 | 40.91 |
| | | (35.75–82.70) | (69.85–97.55) | (53.13–91.51) | (65.52–86.21) | |
| Smear | 36 | 80.00 | 46.15 | 49.51 | 77.76 | 27.78 |
| | | (44.39–97.48) | (26.59–66.63) | (37.95–61.12) | (48.61–92.82) | |

[a]Sn, sensitivity; Sp, specificity; PPV, positive predictive value; and NPV, negative predictive value. Confidence intervals (95% CI) are reported in parentheses. P, prevalence of each subgroup.

cases (N = 10) but was also detected in DfCL (N = 1), CL (N = 2), and one case presenting both MCL and CL (N = 1).

The second most common variant, HSP70-26 (N = 3 samples from 3 patients), was identified in samples from Paraguay and Argentina (Chaco province). A mixed signal was observed in the chromatogram at positions 210 (C/G) and 213 (C/T), suggesting potential heterozygosity at specific nucleotide sites or a coinfection event involving different variants. HSP70-26 was associated with both MCL (N = 2) and CL (N = 1) cases. Based on a Bayesian resolution algorithm, two haplotypes were identified in this variant: HP22 (the same haplotype found in HSP70-24) and HP24.

Four additional variants were identified in *L. (V.) braziliensis* from the patient population (HSP70-60 to 63). HSP70-63, detected in a MCL case from Bolivia, was associated solely with the HP60 haplotype, with no evidence of heterozygosity or coinfection. This haplotype was also present in HSP70-60, in combination with HP22, exhibiting a mixed signal at position 761 (A/G). This variant was found in two patients: one from Argentina (MCL) and another CL case with three possible endemic origins (Bolivia, Brazil, or Paraguay).

HSP70-61 and HSP70-62 were associated with CL cases, one from Argentina and another with an unknown infection origin, both composed of HP22 combined with other haplotypes (HP61 and HP62, respectively).

A haplotype network was constructed to resolve intraspecific relationships in *L. (V.) braziliensis* variants (Fig. 5).

Regarding the haplotypes of *L. (V.) braziliensis* (the most frequent species), HP22 was the most common and geographically widespread, present in all clinical forms described in the study, in accordance with the HSP70-24 variant composed by this haplotype. The haplotypes described in clinical cases from Argentina clustered within this variant, along with HP24, HP60, and HP61, which are closely related in the network. Only HP22, HP24, and HP60 presented more than one clinical manifestation in the cases studied. HP24 was associated with cases of MCL and CL in Paraguay and Argentina, while HP60 was linked to the same clinical forms, but more frequently in Bolivia and to a lesser extent in Argentina.

The combination of five haplotypes resulted in six HSP70 variants within the patient population infected with *L. (V.) braziliensis*. Notably, sample M037 (HSP70-60, HP22, and HP60 at the time of diagnosis) revealed haplotype selection after amphotericin B treatment, with HP60 no longer detected.

Three additional HSP70 variants described in this study correspond to three different *Leishmania* species. HSP70-14, identified in a CL patient from Bolivia, was also found

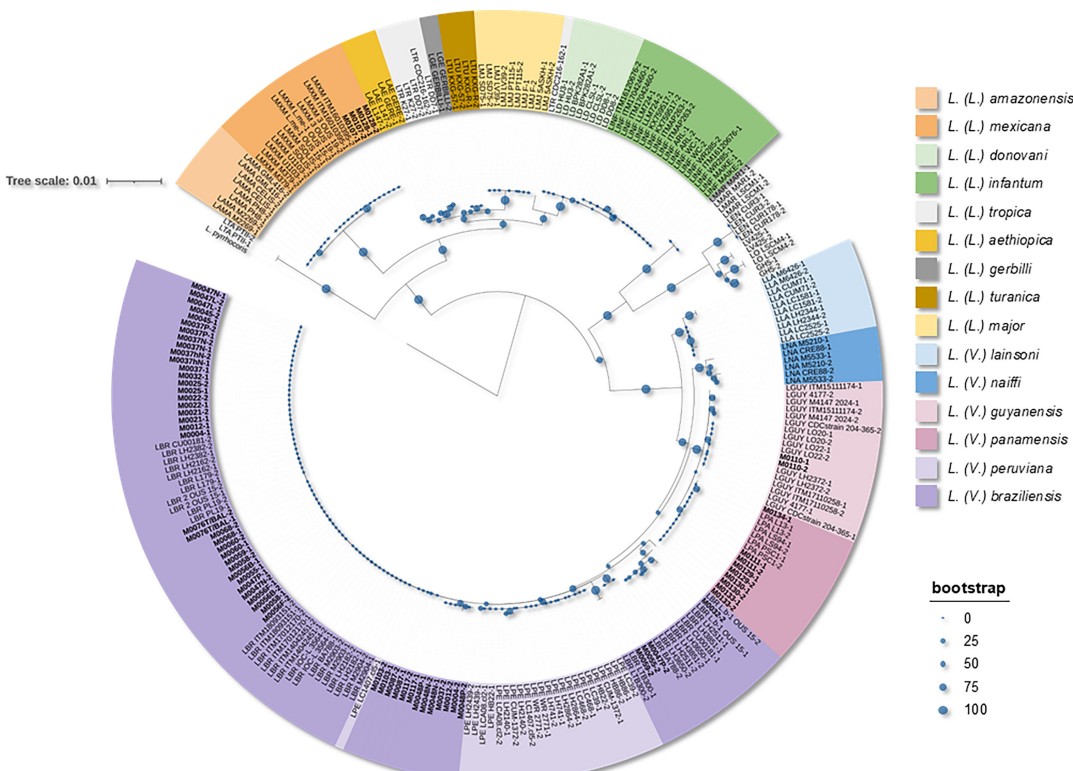

**FIG 4** Phylogenetic analysis of the *hsp*70 marker used for species assignment of clinical samples. A total of 94 reference strains (see Table S2 at https://doi.org/10.5281/zenodo.18854787) were included together with patient-derived sequences (highlighted in bold). Species identification was determined based on the clustering of clinical sequences within well-supported reference clusters.

in four *L. (V.) guyanensis* strains from Bolivia and Peru (see Table S2 at https://doi.org/10.5281/zenodo.18854787). HSP70-18, detected in three CL patients from Costa Rica, Panama, and Argentina, was also reported in *L. (V.) panamensis* strains from Panama (see Table S2 at https://doi.org/10.5281/zenodo.18854787). Finally, HSP70-12, present in six *L. (L.) mexicana* strains from Belize, Guatemala, and Mexico (see Table S2 at https://doi.org/10.5281/zenodo.18854787), was identified in a CL patient from Mexico.

## DISCUSSION

The WHO 2021–2030 roadmap underscores the critical role of accurate leishmaniasis diagnosis in achieving its targets. Although mainly focused on VL, these challenges extend to other clinical forms, where early detection remains essential for effective management. Despite advances, diagnostic tools continue to show limitations in sensitivity, specificity, and standardization, particularly in endemic regions (22). PCR-based methods have shown superior analytical performance, but the lack of validated, standardized protocols has limited their widespread implementation.

This study evaluates a specific PCR protocol targeting the conserved region of the *Leishmania* spp. kDNA minicircle, emphasizing analytical sensitivity and specificity with special attention to the differential diagnosis of *Leishmania* and related pathogens present in endemic areas or associated with clinically similar diseases. The diagnostic performance of the protocol was also assessed.

The most commonly used primers targeting the conserved region of the kDNA minicircle were redesigned to achieve similar Tm and incorporate a single nucleotide polymorphism identified from available sequence data (23–29). Comparison of the two primer sets revealed a clear improvement in specific band amplification, as well as a reduction in nonspecific amplification (data not shown). The analytical specific-ity, assessed through both exclusivity and inclusivity criteria, supports its robustness

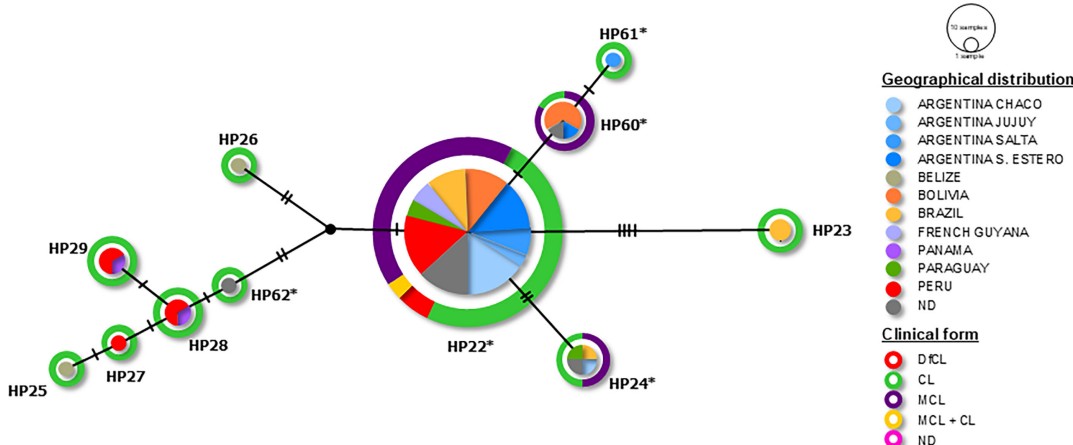

**FIG 5** Haplotype network analysis of the *hsp*70 marker. Median Joining Network of *L. (V.) braziliensis* haplotypes. Geographic distribution is represented as spheres, while clinical forms are depicted as rings. *: Haplotypes (HPs) identified in this study. DfCL, diffuse cutaneous leishmaniasis; CL, cutaneous leishmaniasis; and MCL, mucocutaneous leishmaniasis.

for application in endemic settings. The kDNA-PCR demonstrated high exclusivity, distinguishing *Leishmania* from pathogens with overlapping geographic distribution or clinical profiles. This distinction is particularly relevant in regions where co-endemicity and potential cross-reactivity present diagnostic challenges. Inclusivity tests confirmed the assay's ability to detect a broad range of *Leishmania* species circulating in the region [*L. (V.) braziliensis*, *L. (L.) infantum*, *L. (L.) amazonensis*, *L. (V.) panamensis*, *L. (V.) guyanensis*, and *L. (L.) mexicana*], underscoring its utility for species-level detection across diverse epidemiological settings. Analytical sensitivity assays yielded LoDs of 0.05 parasites/mL for *L. (L.) infantum* and 0.5 parasites/mL for *L. (V.) braziliensis,* demonstrating kDNA-PCR suitability for detecting infections with low parasite burdens. Few studies evaluated LoDs for molecular detection of *Leishmania*, with reported values as low as 1 parasite/mL (30, 31). Notably, this study is the only one to use human blood-derived DNA as a PCR template matrix in parasitemia curves, thereby closely mimicking clinical samples. Moreover, it incorporates an internal amplification control in a multiplex setting.

Diagnostic performance of kDNA-PCR was assessed in 83 patients with clinical suspicion of leishmaniasis. Disease status was determined by smear microscopy (N = 33), which, despite being the reference method, presents several limitations that can affect its reliability (32). Consequently, molecular methods, being inherently more sensitive and specific, may show lower apparent performance due to limited concordance with the reference method, as observed for the PCR under validation.

In this study, 10 patients were positive only by microscopy (12.05%), 16 only by PCR (19.28%), and 23 by both (27.71%). As a result, the estimated prevalence increased from 39.76% (microscopy) to 59.04% when both techniques were considered. Diagnostic sensitivity and specificity of kDNA-PCR reached 69.7% and 68%, respectively; however, these estimates should be interpreted with caution, as they were calculated relative to smear microscopy, an imperfect reference method, and are therefore likely influenced by its limitations. For instance, two false negatives corresponded to *Paracoccidioides* spp. infections, suggesting possible misidentification of fungal elements as amastigotes. Conversely, PCR-only detections may represent true positives undetected by microscopy due to low parasite burdens, which is consistent with previous findings using independent molecular assays (33). Accordingly, the observed PPV (58.98%) and NPV (77.27%) may underestimate true test performance.

In addition, when patients were stratified by the most common clinical forms (CL and MCL), comparable sensitivity and specificity values were observed. However, specificity was lower in MCL cases, likely due to lower parasite loads associated with this clinical form, resulting in a higher proportion of PCR-only detections. Additional evidence

supporting this hypothesis was obtained through amplification and sequencing of the *hsp*70 marker for species identification in eight samples initially classified as "false positives" by the reference method (six from MCL cases and two from CL). Five of these samples were identified as *L. (V.) braziliensis* and three as *L. (V.) panamensis*.

Sampling strategy strongly influenced diagnostic performance. For CL, samples were collected as biopsies from the lesion edge and swabbing the ulcer base, in line with recommendations emphasizing parasite concentration in this region (34). In MCL, where lesions are extensive and less well-defined, swabs improved detection relative to biopsies by sampling a broader area. Consistently, the best agreement with microscopy was observed for CL biopsies (75% sensitivity and 100% specificity) and MCL swabs (82.35% sensitivity and 95% specificity). These findings highlight the importance of tailored sampling strategies according to lesion type and clinical presentation.

The study population also included suspected VL cases. However, due to the limited number of patients (*N* = 11) and samples (*N* = 13), a statistical analysis of this subgroup was not feasible. Within this subset, three false-negative and one false-positive results were observed by kDNA-PCR, based on the same interpretative framework applied to CL and MCL cases.

Species identification remains a critical but often neglected step in the diagnostic process in the Americas. Infection outcome/features and treatment responses are strongly influenced by the parasite species, yet these associations are still evolving (35).

While the conserved kDNA region provides excellent analytical sensitivity, it lacks discriminatory power for species identification.

On the other hand, the *hsp*70 gene remains the most validated marker for this purpose (16, 18), fulfilling key requirements of reproducibility and broad applicability, but its lower sensitivity (≈300 parasites/mL) limits its use as a primary diagnostic tool. Based on this rationale, the proposed diagnostic workflow integrates a highly sensitive detection step (kDNA-PCR), essential for early treatment initiation, followed by species characterization using the globally validated *hsp*70 gene marker.

In this study, the original *hsp*70 primer set was optimized, allowing successful identification of four *Leishmania* species in clinical samples: *L. (L.) mexicana*, *L. (V.) guyanensis*, *L. (V.) panamensis*, and *L. (V.) braziliensis*, the latter being the most frequent.

A total of 35 samples from 26 patients were successfully characterized, regardless of sample type. Consistent with kDNA-PCR findings, species identification was more frequent from mucosal swabs, reinforcing their use as a minimally invasive and effective method, especially in MCL cases with low parasite burdens.

Among autochthonous Argentinean cases, only *L. (V.) braziliensis* and *L. (V.) panamensis* were detected. *L. (V.) braziliensis* remains the main species associated with CL and MCL, while *L. (V.) panamensis* was identified for the second time in the country, previously reported in a CL patient from Salta Province (11). These findings underscore the importance of identifying the circulating species in each region rather than assuming the predominance of commonly reported species. The introduction of new species through travel-related cases underscores the need for continuous species-level surveillance. Moreover, one of the greatest challenges for leishmaniasis control programs is the broad clinical spectrum induced by *Leishmania* infection, which cannot be reliably predicted based on species alone. To date, no comprehensive studies on species distribution have been conducted in Argentina. Therefore, this study represents a significant initial step toward filling this knowledge gap and supporting informed medical decision-making through a better understanding of national species diversity.

Finally, the analysis of the *hsp*70 marker enabled the evaluation of intraspecific variants, revealing potentially distinct dynamics in clinical cases from which multiple samples were collected either at diagnosis or along diagnosis and post-treatment. In patient LIBEI_13 (see Table S1 at https://doi.org/10.5281/zenodo.18854787), a variant shift was detected after treatment with amphotericin B: variant HSP70-60 was identified at diagnosis, whereas variant HSP70-24 was detected after treatment. Both shared haplotype HP22; however, HP60 was not detected after treatment. These findings may

suggest a therapy-induced selection of parasite subpopulations, although further studies are needed to confirm this observation. In patient LIBEI_18 (see Table S1 at https://doi.org/10.5281/zenodo.18854787), a MCL case from Chaco, Argentina, different mucosal sites harbored distinct variants. HSP70-24 (HP22) was found in both nasal and laryngeal samples, whereas the palate sample revealed variant HSP70-26, composed of haplotypes HP22 and HP24, as defined by two heterozygous sites in the sequences. This may reflect a within-host heterogeneous distribution of parasite populations, which could have implications for treatment evaluation, although this hypothesis requires further investigation.

However, when interpreting our results, it should be considered that the analysis was conducted at a single center, which may limit the generalizability of the findings to other endemic settings. Second, the retrospective design relied on previously collected clinical samples and records, potentially introducing selection bias. Finally, although external validation in independent cohorts from different geographic regions was not performed in the present study, this protocol is currently being extended to additional centers, and such validation will be essential to confirm the robustness and broader applicability of the proposed diagnostic algorithm. In this context, future analyses of larger, multicenter data sets could benefit from the application of latent class models, which may allow a more accurate estimation of the true diagnostic performance of the molecular assay in the absence of a perfect reference standard.

The diagnostic workflow evaluated in this study provides a robust and flexible framework for detecting and characterizing *Leishmania* spp., addressing key limitations of current diagnostic approaches. The combined use of kDNA-PCR for sensitive detection and *hsp*70-based sequencing for species identification constitutes a powerful tool for clinical decision-making and epidemiological surveillance. Importantly, the limited accuracy of the current WHO reference method highlights the urgent need for improved diagnostic algorithms. In this context, the proposed workflow represents a significant step forward, offering enhanced sensitivity, specificity, and species-level resolution.

Moreover, integrating serological testing into this workflow could further enhance diagnostic accuracy, particularly in VL cases or when sample availability is limited. An integrated diagnostic approach, combining molecular, serological, and clinical data, would not only improve individual case resolution but also strengthen disease control strategies in both endemic and non-endemic regions (36).

Collectively, these findings underscore the importance of implementing comprehensive and standardized diagnostic protocols capable of capturing the full clinical and epidemiological spectrum of *Leishmania* infections.

## MATERIALS AND METHODS

### Reference strains and culture conditions

The diagnostic PCR was standardized using DNA from reference strains of Argentina relevant species, *L. (L.) infantum* (MHOM/BL/67/ITMAP263), *L. (L.) amazonensis* (IFLA/BR/1967/PH8), and *L. (V.) braziliensis* (MHOM/BR/75/M2903). The strains were cultured at 23°C in complete RPMI 1640 (supplemented with 0.1 mM sodium pyruvate, 2 mM L-glutamine, 100 U/mL penicillin, 100 µg/mL streptomycin, and 10% fetal bovine serum) and USMARU medium (Difco blood agar medium with 20% defibrinated rabbit blood).

### Parasite DNA extraction

In each case, $1 \times 10^5$ promastigotes were harvested, and DNA was extracted using the DNA Puriprep S-Kit (INBIO Highway, Argentina), following the manufacturer's instructions for cultured cells.

## Optimization of amplification conditions

PCR targeting a 120 bp fragment of the conserved region of kDNA minicircles was optimized using primers kDNA22-FN (5′-KAGGGGCGTTCTSCGAAAW-3′) and kDNA22-RN (5′-SSSKCTATWTTACACCAACCCC-3′) modified from Rodgers et al. (19). The reaction mixture consisted of 4 µL of DNA from reference strains of *Leishmania* spp. in a total volume of 25 µL, containing 1 U of T-Holmes DNA polymerase (INBIO Highway, Argentina), Mint Buffer (1×), 200 µM of each dNTP, and variable concentrations of $MgCl_2$ (1.5, 2, or 2.5 mM) and primers (0.2, 0.3, 0.4, or 0.5 µM). Cycling conditions followed the protocol optimized by Almazán et al. (23), which amplifies the conserved region of kDNA minicircles. A temperature gradient (51°C–63°C) was used to determine the optimal annealing temperature. The final amplification protocol consisted of an initial denaturation at 94°C for 1 min, followed by 40 cycles of 94°C for 20 s, annealing at an optimized temperature for 1 min and 72°C for 1 min, without a final extension step. Amplifications were performed using an Applied Biosystems Veriti 96-Well Thermal Cycler (Thermo Fisher Scientific Inc., MA, USA). The PCR was optimized in a multiplex format by including primers for the human *β-globin* gene as an internal amplification control (37).

DNA extracted from reference *Leishmania* spp. strains (positive control) and human blood DNA (internal control) were used to validate the PCR assay.

Amplification products were visualized on 3% agarose gels stained with GelRed (Biotium, CA, USA) using a Fotodyne Foto/UV 21 Transilluminator (Thermo Fisher Scientific Inc., MA, USA).

## kDNA-PCR specificity analysis

The specificity of the kDNA-PCR assay was evaluated using isolates from patients with fungal infections that cause cutaneous or mucosal lesions and share endemic areas with *Leishmania* spp. (38). These included endemic mycoses such as *Histoplasma capsulatum*, *Paracoccidioides brasiliensis*, *Sporothrix* spp., and *Candida* spp. Fungal DNA presence was confirmed by a PCR targeting the conserved fungal ribosomal Internal Transcribed Spacer following the protocol reported by Elías et al. (38).

As endemic areas of Chagas disease and leishmaniasis overlap, and considering the phylogenetic proximity of *Trypanosoma cruzi* to the *Leishmania* genus, potential cross-reactivity was evaluated. kDNA-PCR specificity testing included DNA from *T. cruzi* (250 parasites/mL RA strain, DTU VI) as template. To confirm *T. cruzi* DNA presence, a specific PCR was performed to amplify a nuclear minisatellite region of 188 bp called TCZ (39).

Additionally, DNA extracted from human blood was tested with kDNA22-FN and kDNA22-RN primers to complete the specificity analysis of the kDNA-PCR assay. PCR amplification products were visualized as mentioned in the section "Optimization of amplification conditions."

## Detection limit analysis for kDNA-PCR

To determine the limit of detection (LoD), DNA dilution curves were prepared from reference strains mentioned in the section "Reference strains and culture conditions" at known promastigote concentrations in the presence of human blood DNA. Briefly, 200 µL of human blood was artificially contaminated with $1 \times 10^5$ parasites/mL, and DNA was extracted using the DNA Puriprep S-Kit (INBIO Highway, Argentina), following the manufacturer's instructions for blood samples. Serial dilutions of the extracted DNA were then prepared in DNA extracted from uninfected human blood to achieve DNA equivalent to 1, 0.5, 0.1, 0.05, and 0.01 parasites/mL. DNA concentration and quality were assessed using a Nano-500 Micro-Spectrophotometer (Hangzhou Allsheng Instruments Co., Ltd) to ensure consistency and DNA integrity.

The LoD was defined according to the criteria proposed by Vaks et al., identifying the lowest parasite concentration consistently detectable (>95%) by the optimized kDNA-PCR (40).

## Patient population and clinical samples

This was a cross-sectional and descriptive study conducted between 2015 and 2024 at the Infectious Diseases Division, Hospital de Clínicas "José de San Martín," Universidad de Buenos Aires (City of Buenos Aires, Argentina). A total of 85 patients over 18 years of age who presented with clinical suspicion of leishmaniasis were included. All participants met predefined inclusion criteria: history of residence or travel within the previous year to an endemic or transmission area with proven vector presence and compatible clinical manifestations. These comprised cutaneous or mucocutaneous lesions suggestive of leishmaniasis, or, in the case of suspected VL, fever persisting for more than 2 weeks. Patients were excluded if they had received antileishmanial treatment within the last 3 months, were receiving experimental drugs, or were neonates or infants.

Clinical samples ($N$ = 152) from these patients were stored at the biobank of the Biomolecular Research Laboratory on Infectious Diseases (IMTIB, CONICET-HIBA-UHIBA) and analyzed for this study (see Table S1 at https://doi.org/10.5281/zenodo.18854787).

A sample was considered positive for *Leishmania* spp. when amastigotes were detected by microscopy in smears stained using the Tinción 15 Kit (Biopur SRL, Argentina) and/or when the kDNA-PCR yielded a positive result. Samples with discordant results (microscopy negative/PCR positive) were further analyzed by *hsp*70-PCR and sequencing. Cases confirmed by sequencing were classified as true PCR positives.

For each case (confirmed positive or negative), information on endemic area history for leishmaniasis (exposure risk) and clinical manifestations determined by the Ministry of Health of Argentina was available (41).

Samples from patient follow-up ($N$ = 21) were included to evaluate the diagnostic performance of kDNA-PCR by sample format (biopsy, smear, or swab).

The biobank samples consist of biopsies, smears, or swabs from skin or mucosal lesions, or aliquots from bone marrow aspirates. It should be noted that the latter were not specifically collected for this study but rather correspond to aliquots obtained at healthcare centers during initial diagnosis.

One cerebrospinal fluid (CSF) sample and two bronchoalveolar lavage (BAL) samples obtained at the time of diagnosis, as well as one urine sample and one nasal secretion (NS) sample, were tested.

## DNA extraction from clinical samples

The DNA from the samples was extracted using the DNA Puriprep S-Kit (INBIO Highway, Argentina) for swabs, scrapings, urine, NS, BM, BAL, and CSF, or the DNA PuriPrep-T Kit (INBIO Highway) for tissue samples (biopsies and fixed tissues), following the manufacturer's instructions.

## Diagnostic performance of kDNA-PCR and statistical analysis

Sensitivity, specificity, PPV, and NPV of kDNA-PCR were expressed as percentages and calculated using microscopy as the reference method. Analysis was performed per patient and per sample type. Confidence intervals for sensitivity and specificity are "exact" Clopper-Pearson confidence intervals (95%). Confidence intervals for the predictive values were the standard logit confidence intervals given by Mercaldo et al. except when the predictive value was 0% or 100% (Clopper-Pearson confidence interval) (42).

## Determination of *Leishmania* spp. using the *hsp*70 marker

All clinical samples that tested positive by kDNA-PCR were subjected to PCR for a 1,245 bp region of the *hsp*70 gene to determine the infecting species. This includes two

patients (LIBEI_4 and LIBEI_12) and their two respective samples (M012 and M032) that were not tested as described in section "Diagnostic performance of kDNA-PCR and statistical analysis," since they were not diagnosed using the reference method but only by kDNA-PCR.

The reaction mixture consisted of 4 µL of template DNA in a total volume of 25 µL, with final concentrations of 1× reaction buffer (INBIO Highway, Argentina), 1 U of Typhon DNA polymerase (INBIO Highway, Argentina), 200 µM of each dNTP, 0.8 µM of each primer (F25: 5′-GGACGCCGGCACGATTKCT-3′ and R1310.N: 5′-CCTGGTTGYTGYT CAGCCACTC-3′), and 0.1 mg/mL of enhancer Typhon (INBIO Highway, Argentina). The amplification cycles were: 98°C for 30 s, followed by 30 cycles of 98°C for 10 s, 70°C for 30 s, and 72°C for 50 s, with a final extension at 72°C for 10 min using an Applied Biosystems Veriti 96-Well Thermal Cycler (Thermo Fisher Scientific Inc., MA, USA). The amplification products were visualized on 2% agarose gels stained with GelRed (Biotium, USA) using a Fotodyne Foto/UV 21 Transilluminator (Thermo Fisher Scientific Inc., MA, USA). Amplification products were sequenced by Sanger sequencing at Macrogen Inc. (Seoul, South Korea).

The consensus sequences were generated by assembling forward and reverse strands using the STADEN Package software (MRC-LMB, UK). We also visually inspected both strands for identifying ambiguous sites when two peaks overlapped in both chromatograms. The consensus sequences were aligned and trimmed to a length of 1,116 bp in frame using MEGA11 software (43). Haplotype reconstruction was carried out with DNAsp version 6 using the PHASE algorithm, which considers the entire haplotype population (44).

Sequences obtained from the patient population were further analyzed alongside 94 worldwide sequences sourced from the GenBank database (see Table S2 at https://doi.org/10.5281/zenodo.18854787). HSP70 variants and haplotypes coding was performed using MLSTest version 1.0.1.23 software (45).

Phylogenetic relationships were inferred by the Neighbor Joining method, tested with 500 bootstrap replications using MLSTest software. Ambiguous data were treated as average states to avoid misinterpretations. Genealogical associations were studied by constructing a haplotype network using the Median-Joining algorithm (epsilon = 0) with PopART version 1.7 software (46).

## ACKNOWLEDGMENTS

This work received support from Agencia I+D+i (PICT 2021-GRF-TII-00216). The financial support was exclusively allocated to laboratory reagents and experimental work. The sequencing of reference strains used in this study was conducted during J.J.L.'s Ph.D. at Kochi Medical School (MEXT scholarship program, Kochi University, Japan) and supported by the Institute of Tropical Medicine (NEKKEN), Nagasaki University (research grant No. 2019-Ippan-3). The funders had no role in study design, data analysis, or publication decisions.

We thank the Divisions of Otolaryngology, Dermatology, Infectious Diseases, and Internal Medicine at Hospital de Clínicas "José de San Martín" (UBA) and the Parasitology and Travel Medicine Sections at Hospital de Infecciosas Francisco Javier Muñiz for their collaboration in clinical case discussions. We also acknowledge the Mycology Center, the Trypanosomatid Biochemistry and Cell Biology and LAVAX laboratories (IMPaM, UBA–CONICET), and the Instituto de Investigaciones Biomédicas en Retrovirus y Sida (UBA–CONICET) for their support. We are grateful to Dr. Masataka Korenaga for his encouragement during the leishmaniasis research.

M.G.R., S.A.R., and P.R. conceptualized the study. M.G.R., W.L.Q.-G., S.A.R., and P.R. curated the data. M.G.R., S.A.R., and P.R. performed formal analysis. P.R. acquired funding. M.G.R., W.L.Q.-G., J.J.L., G.S., C.P., C.G., G.A., S.E., M.B.F., A.A.P., S.A.R., and P.R. performed the investigation. M.G.R., W.L.Q.-G., J.J.L., G.S., C.P., S.A.R., and P.R. designed the methodology. M.G.R. and P.R. contributed to project administration, supervised the study, visualized the study, and wrote the original draft. C.G., G.A., S.E., M.B.F., A.A.P., and S.A.R. provided

resources. M.G.R., W.L.Q.-G., and P.R. validated the study. M.G.R., W.L.Q.-G., J.J.L., S.A.R., and P.R. reviewed and edited the manuscript. All authors gave final approval of the version to be published.

## AUTHOR AFFILIATIONS

[1]Instituto de Medicina Traslacional e Ingeniería Biomédica, Consejo Nacional de Investigaciones Científicas y Técnicas, Universidad Hospital Italiano, Hospital Italiano de Buenos Aires, Ciudad Autónoma de Buenos Aires, Argentina
[2]Hospital de Infecciosas "Francisco Javier Muñiz", Ciudad Autónoma de Buenos Aires, Argentina
[3]Hospital de Clínicas "José de San Martín", Universidad de Buenos Aires, Ciudad Autónoma de Buenos Aires, Argentina
[4]Instituto de Investigaciones en Microbiología y Parasitología Médica, Universidad de Buenos Aires, Consejo Nacional de Investigaciones Científicas y Técnicas, Ciudad Autónoma de Buenos Aires, Argentina

## AUTHOR ORCIDs

Paula Ruybal http://orcid.org/0000-0002-2365-7000

## FUNDING

| Funder | Grant(s) | Author(s) |
| --- | --- | --- |
| Agencia Nacional de Promoción de la Investigación, el Desarrollo Tecnológico y la Innovación | PICT 2021-GRF-TII-00216 | Marikena Guadalupe Risso |
| | | Juan Jose Lauthier |
| | | Silvia Analia Repetto |
| | | Paula Ruybal |
| Nagasaki University | 2019-Ippan-3 | Juan Jose Lauthier |

## AUTHOR CONTRIBUTIONS

Marikena Guadalupe Risso, Conceptualization, Data curation, Formal analysis, Investigation, Methodology, Project administration, Supervision, Validation, Visualization, Writing – original draft | Wendy Lorena Quintero-Garcia, Data curation, Investigation, Methodology, Validation, Writing – review and editing | Juan Jose Lauthier, Investigation, Methodology, Writing – review and editing | Giuliana Saggion, Investigation, Methodology | Carlos Parra, Investigation, Methodology | Catalina Gauder, Investigation, Resources | German Astudillo, Investigation, Resources | Sofia Echazarreta, Resources | Monica Beatriz Foccoli, Resources | Ana Andrea Pisarevsky, Resources | Silvia Analia Repetto, Conceptualization, Data curation, Formal analysis, Investigation, Resources, Writing – review and editing | Paula Ruybal, Conceptualization, Data curation, Formal analysis, Funding acquisition, Investigation, Methodology, Project administration, Supervision, Validation, Visualization, Writing – original draft

## DATA AVAILABILITY

The hsp70 gene sequences generated de novo in this study have been deposited in the NCBI GenBank database (https://www.ncbi.nlm.nih.gov/nucleotide/) under accession numbers PX241371–PX241405.

## ETHICS APPROVAL

The study was approved by the Ethics Committees of the Facultad de Medicina, Universidad de Buenos Aires (RESCD-2023-360-E-UBA-DCT#FMED) and the Hospital de Clínicas "José de San Martín" (RTO-UBA: 0396919/2015) and conducted according to the

ethical principles of the Declaration of Helsinki for medical research involving human subjects.

## ADDITIONAL FILES

The following material is available online.

## Open Peer Review

**PEER REVIEW HISTORY (review-history.pdf).** An accounting of the reviewer comments and feedback.

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
