## [Reviewer comments · Microbiology Spectrum]

Microbiology Spectrum

An integrated molecular approach for the detection and species-level identification of *Leishmania* spp. in clinical samples

Marikena Risso, Wendy Quintero Garcia, Juan Lauthier, Giuliana Saggion, Carlos Parra, Catalina Gauder, Germán Astudillo, Sofía Echazarreta, Monica Foccoli, Ana Pisarevsky, Silvia Repetto, and Paula Ruybal

Corresponding Author(s): Paula Ruybal, Instituto de Medicina Traslacional e Ingeniería Biomédica (CONICET-HIBA-UHI)

Review Timeline:

Submission Date:	November 17, 2025
Editorial Decision:	January 26, 2026
Revision Received:	January 29, 2026
Accepted:	February 26, 2026

Editor: Ana Cabrera

Reviewer(s): Disclosure of reviewer identity is with reference to reviewer comments included in decision letter(s). The following individuals involved in review of your submission have agreed to reveal their identity: Estefania Calvo Alvarez (Reviewer #1); CLAUDIO Vieira da Silva (Reviewer #2)

Transaction Report:

DOI: <https://doi.org/10.1128/spectrum.03665-25>

Re: Spectrum03665-25 (An integrated molecular approach for the detection and species-level identification of *Leishmania* spp. in clinical samples)

Dear Dr. Paula Ruybal:

Thank you for the privilege of reviewing your work. Below you will find my comments, instructions from the Spectrum editorial office, and the reviewer comments.

Revision Guidelines

Sincerely,
Ana Cabrera
Editor
Microbiology Spectrum

Reviewer #2 (Comments for the Author):

This manuscript addresses an important question in public health within the Americas, specifically in Argentina: the necessity for a diagnostic tool that not only detects the parasite but also identifies the *Leishmania* species. Given that different species (*L. braziliensis* vs. *L. infantum*) require distinct therapeutic approaches and present varying prognoses, the clinical relevance of this work is indisputable and timely. While the use of kDNA for detection and hsp70 for typing is not a novel biological discovery, the novelty lies in the standardization of an integrated and optimized molecular workflow tailored to regional clinical realities. The redesign of kDNA22 primers to balance annealing temperatures (T_m) and the validation of swabs as a minimally invasive

collection method for mucocutaneous leishmaniasis (MCL) represents significant incremental technical advances. Technical optimization of primers; validation of swabs for MCL; in-depth analysis of intraspecific variants of *L. braziliensis* highlights the work. However, the sample size for Visceral Leishmaniasis (VL) cases is small (N=11), limiting definitive conclusions for this specific clinical form. Minor adjustments: In the discussion, clarify whether the low specificity (68%) compared to microscopy could be further analyzed using a latent class model to better reflect the true performance of the molecular test and ensure species nomenclature is strictly consistent with the latest taxonomic guidelines mentioned in the introduction (Akhoundi et al.).

This manuscript describes an integrated molecular approach combining kDNA-PCR for detection of *Leishmania* spp with *hsp70*-based sequencing for species identification in clinical samples. The experimental work is technically sound, well executed and relevant for *Leishmania*-endemic and resource-limited settings. The inclusion of species typing and haplotype analyses is a strong aspect of the study. However, several important issues need to be addressed to strengthen the quality of the manuscript.

Major comments

1. The Results state that four *Leishmania* species were identified among the clinical samples based on *hsp70* analysis, with specific numbers reported for each species. However, there is currently no figure or supplementary material that explicitly supports this statement. Figure 4 presents a phylogenetic tree, but it does not clearly delineate species boundaries, nor does it allow the reader to independently verify species assignments or the reported sample counts. As a result, the link between the phylogenetic analysis and the numerical species distribution described in the text is not clear. The authors should provide an explicit visual or tabular summary (main or supplementary) linking each clinical sample to its assigned species. In addition, including a supplementary figure showing an alignment of the *hsp70* sequences used for species identification (highlighting diagnostic positions) would strengthen the support for the reported species assignments.

2. The manuscript states that “a total of nine *hsp70* variants were identified among the patient-derived sequences,” referring to Figure 4. However, it is not clear how this number can be inferred from the figure as currently presented. Figure 4 shows phylogenetic relationships but does not explicitly label variants or indicate how variants are defined and counted. The authors should clarify whether the identification of *hsp70* variants derives from sequence/haplotype analyses rather than directly from the phylogenetic tree. Please, revise the text accordingly or modify the figure/legend to make this information explicit.

3. The manuscript does not clearly specify which *hsp70* gene type is being amplified (*hsp70-I* or *hsp70-II*), nor how primer design relates to known differences in HSP70 locus organization among *Leishmania* species. Previous work has demonstrated that the genomic organization of the HSP70 locus differs across *Leishmania* species, including the absence of the *hsp70-II* gene in *L. braziliensis* (Folgueira et al., Parasitology, 2007, doi: 10.1017/S0031182006001570). Without clarification of which gene copy or region is targeted, it is difficult to fully interpret sequence diversity, species discrimination and variant analyses. The authors should clearly state which *hsp70* gene type is amplified, which sequence was used for primer design and how differences in HSP70 locus organization across species were considered.

4. Given the central role of *hsp70* in this study, the Introduction and/or Discussion should acknowledge pioneering work that established HSP70 as a molecular marker for species identification in *Leishmania*. As previously mentioned, the study by Folgueira et al. (Parasitology, 2007), which describes the genomic organization of the HSP70 locus and its implications for species discrimination, is directly relevant to the present work. In addition, the usefulness of the *hsp70* 3'-UTR region for species identification,

as demonstrated by Requena et al. (Parasit Vectors, 2012, doi: 10.1186/1756-3305-5-87), should be commented on even if the authors did not specifically target this region.

Minor comments

1. The interpretation of diagnostic sensitivity and specificity should be presented more cautiously, as these values are calculated relative to smear microscopy, which is an imperfect reference standard.
2. The limitations of the study (single-center design, retrospective analysis, lack of external validation) should be stated in the Discussion.
3. In the Discussion, the gene *hsp70* should be consistently italicized.

**An integrated molecular approach for the detection and species-level identification of
Leishmania spp. in clinical samples**

Marikena Guadalupe Risso, Wendy Lorena Quintero-Garcia, Juan Jose Lauthier, Giuliana Saggion, Carlos Parra, Catalina Gauder, German Astudillo, Sofia Echazarreta, Monica Beatriz Foccoli, Ana Andrea Pisarevsky, Silvia Analia Repetto, Paula Ruybal

RESPONSE TO REVIEWERS' COMMENTS

Comments for the Author

Reviewer #1:

This manuscript describes an integrated molecular approach combining kDNA-PCR for detection of *Leishmania* spp with *hsp70*-based sequencing for species identification in clinical samples. The experimental work is technically sound, well executed and relevant for *Leishmania*-endemic and resource-limited settings. The inclusion of species typing and haplotype analyses is a strong aspect of the study. However, several important issues need to be addressed to strengthen the quality of the manuscript.

Major comments

Comment 1:

The Results state that four *Leishmania* species were identified among the clinical samples based on *hsp70* analysis, with specific numbers reported for each species. However, there is currently no figure or supplementary material that explicitly supports this statement. Figure 4 presents a phylogenetic tree, but it does not clearly delineate species boundaries, nor does it allow the reader to independently verify species assignments or the reported sample counts. As a result, the link between the phylogenetic analysis and the numerical species distribution described in the text is not clear. The authors should provide an explicit visual or tabular summary (main or supplementary) linking each clinical sample to its assigned species. In addition, including a supplementary figure showing an alignment of the *hsp70* sequences used for species identification (highlighting diagnostic positions) would strengthen the support for the reported species assignments.

Response:

We thank the reviewer for this helpful and constructive comment. To enhance clarity and to explicitly link the phylogenetic analysis with species assignments and sample counts, we have now directly referenced Supplementary Table 1 in the Results section describing the *hsp70*-based species typing (line 269, marked-up manuscript file). This table provides a detailed summary of each clinical sample, its corresponding *hsp70* sequence variant, and the assigned *Leishmania* species derived from the phylogenetic analysis, thereby allowing independent verification of the reported species distribution.

Figure 4 continues to illustrate the evolutionary relationships among clinical samples and global reference strains, while Supplementary Table 1 now clearly supports the numerical species distribution reported in the text.

We also appreciate the suggestion to include a multiple sequence alignment highlighting diagnostic positions, as this could indeed be informative in some contexts. However, the *hsp70* marker used for

Leishmania species identification has been extensively validated and is widely accepted as a robust typing tool, as documented in the literature cited throughout the manuscript. We consider that, in this study, species assignments are supported by well-resolved phylogenetic analyses and by the explicit correspondence between samples and species provided in Supplementary Table 1.

Comment 2:

The manuscript states that “a total of nine *hsp70* variants were identified among the patient-derived sequences,” referring to Figure 4. However, it is not clear how this number can be inferred from the figure as currently presented. Figure 4 shows phylogenetic relationships but does not explicitly label variants or indicate how variants are defined and counted. The authors should clarify whether the identification of *hsp70* variants derives from sequence/haplotype analyses rather than directly from the phylogenetic tree. Please, revise the text accordingly or modify the figure/legend to make this information explicit.

Response:

We appreciate the reviewer’s observation and apologize for the lack of clarity in the original presentation. The total number of nine *hsp70* variants was determined based on sequence differences identified among the patient-derived *hsp70* sequences. These variants correspond to distinct sequence types that do not overlap between species in our reference dataset and therefore can be used as molecular signatures.

Species assignment was subsequently established through phylogenetic analysis, in which both known and newly identified variants were assigned according to their clustering with validated reference sequences. In cases where novel variants were detected, species identification relied on their phylogenetic placement within well-supported clusters.

To avoid ambiguity, we have revised the Results section and the legend of Figure 4 to explicitly distinguish between variant definition based on sequence analysis and species assignment based on phylogenetic clustering (lines 826 to 829, marked-up manuscript file).

Comment 3:

The manuscript does not clearly specify which *hsp70* gene type is being amplified (*hsp70-I* or *hsp70-II*), nor how primer design relates to known differences in HSP70 locus organization among *Leishmania* species. Previous work has demonstrated that the genomic organization of the HSP70 locus differs across *Leishmania* species, including the absence of the *hsp70-II* gene in *L. braziliensis* (Folgueira et al., Parasitology, 2007, doi: 10.1017/S0031182006001570). Without clarification of which gene copy or region is targeted, it is difficult to fully interpret sequence diversity, species discrimination and variant analyses. The authors should clearly state which *hsp70* gene type is amplified, which sequence was used for primer design and how differences in HSP70 locus organization across species were considered.

Response:

We thank the reviewer for highlighting this important point. The *hsp70* typing strategy used in this study specifically targets the *hsp70-I* gene copy, which is conserved across *Leishmania* species and has been widely employed for species identification and phylogenetic analyses.

We have now clarified in the Introduction (line 113, marked-up manuscript file) and Results (lines 255 to 262, marked-up manuscript file) sections that the primers used in this study were originally described in previous publications targeting the *hsp70* type I locus (Van der Auwera G, 2013, <https://doi.org/10.3390/pathogens13010019>). In our work, we did not design primers *de novo*; rather, a modified reverse primer was introduced based on published *hsp70* type I sequences to optimize amplification across diverse species (lines 256 to 259, marked-up manuscript file).

In addition, we have explicitly stated that this *locus* was selected to account for known differences in HSP70 genomic organization among *Leishmania* species, including the absence of the *hsp70* type II gene in *L. (V.) braziliensis*, as previously described.

Comment 4:

Given the central role of *hsp70* in this study, the Introduction and/or Discussion should acknowledge pioneering work that established HSP70 as a molecular marker for species identification in *Leishmania*. As previously mentioned, the study by Folgueira et al. (Parasitology, 2007), which describes the genomic organization of the HSP70 locus and its implications for species discrimination, is directly relevant to the present work. In addition, the usefulness of the *hsp70* 3'-UTR region for species identification, as demonstrated by Requena et al. (Parasit Vectors, 2012, doi: 10.1186/1756-3305-5-87), should be commented on even if the authors did not specifically target this region.

Response:

We thank the reviewer for this important suggestion. We have expanded both the Introduction (line 113, marked-up manuscript file) and the Results (lines 259 to 262, marked-up manuscript file) sections to acknowledge pioneering studies that established *hsp70* as a robust molecular marker for *Leishmania* species identification. In particular, we have incorporated the work by Folgueira et al. (Parasitology, 2007), which describes the genomic organization of the *hsp70* locus and its relevance for species discrimination, as well as the study by Requena et al. (Parasitology Vectors, 2012), highlighting the diagnostic potential of the *hsp70* 3'-untranslated region, even though it was not directly analyzed in this study.

The marker used in the present study corresponds to the *hsp70* type I coding region, originally described and validated by Van der Auwera et al. (2013, <https://doi.org/10.3390/pathogens13010019>), which was selected due to the extensive availability of reference sequences and its proven performance for reliable species assignment across multiple clinical settings. Nevertheless, the inclusion of the above-mentioned references provides important biological and methodological context supporting the broader utility of *hsp70*-based approaches for *Leishmania* typing.

Minor comments

Comment 1:

The interpretation of diagnostic sensitivity and specificity should be presented more cautiously, as these values are calculated relative to smear microscopy, which is an imperfect reference standard.

Response:

We agree with the reviewer and have revised the Discussion to present the interpretation of diagnostic sensitivity and specificity more cautiously. We now explicitly emphasize that these estimates were calculated using smear microscopy as the reference method, which is known to be imperfect. As a result, we now more explicitly acknowledge that the limitations of microscopy, particularly its reduced sensitivity, may lead to an underestimation of the true diagnostic performance of kDNA-PCR (lines 370-372, marked-up manuscript file).

Comment 2:

The limitations of the study (single-center design, retrospective analysis, lack of external validation) should be stated in the Discussion.

Response:

We thank the reviewer for this suggestion. We have now included a dedicated paragraph in the Discussion acknowledging the main limitations of the study, including its single-center design, retrospective nature, and the absence of external validation, and we briefly discuss how these factors may affect the generalizability of the findings (lines 442 to 448, marked-up manuscript file).

Comment 3:

In the Discussion, the gene *hsp70* should be consistently italicized.

Response:

We have revised the Discussion section to ensure that the gene name *hsp70* is consistently italicized throughout the manuscript, in accordance with standard nomenclature conventions (lines 403 and 408, marked-up manuscript file).

Reviewer #2:**Comment:**

This manuscript addresses an important question in public health within the Americas, specifically in Argentina: the necessity for a diagnostic tool that not only detects the parasite but also identifies the *Leishmania* species. Given that different species (*L. braziliensis* vs. *L. infantum*) require distinct therapeutic approaches and present varying prognoses, the clinical relevance of this work is indisputable and timely. While the use of kDNA for detection and *hsp70* for typing is not a novel biological discovery, the novelty lies in the standardization of an integrated and optimized molecular workflow tailored to regional clinical realities. The redesign of kDNA22 primers to balance annealing temperatures (T_m) and the validation of swabs as a minimally invasive collection method for mucocutaneous leishmaniasis (MCL) represents significant incremental technical advances. Technical optimization of primers; validation of swabs for MCL; in-depth analysis of intraspecific variants of *L. braziliensis* highlights the work. However, the sample size for Visceral Leishmaniasis (VL) cases is small ($N=11$), limiting definitive conclusions for this specific clinical form.

Minor adjustments: In the discussion, clarify whether the low specificity (68%) compared to microscopy could be further analyzed using a latent class model to better reflect the true performance of the molecular test and ensure species nomenclature is strictly consistent with the latest taxonomic guidelines mentioned in the introduction (Akhoundi et al.).

Response:

We sincerely thank the reviewer for the thorough evaluation of our manuscript and for the positive and constructive comments. We greatly appreciate the recognition of the clinical relevance of this work, as well as the assessment of the technical contributions related to primer optimization, validation of swab sampling for MCL, and the analysis of intraspecific variability in *Leishmania (Viannia) braziliensis*.

Given the limited number of Visceral Leishmaniasis (VL) cases, we fully agree that this poses a constraint on drawing definitive conclusions about this clinical form. As explicitly stated in the revised Discussion section, we are currently expanding this study to multiple centers, which will allow the inclusion of a larger and more diverse cohort. These additional data will also enable a more robust and unbiased evaluation of diagnostic performance.

We appreciate the reviewer's valuable suggestion concerning the potential use of a latent class model to further analyze the observed low specificity compared to microscopy and to better estimate the true performance of the molecular assay. This approach represents an important methodological improvement, and we plan to implement this analysis once the expanded multicenter dataset

becomes available. We incorporated a sentence stating this to the Discussion section and are grateful for this insightful recommendation, which will directly inform the next phase of our work (lines 442 to 448, marked-up manuscript file).

Finally, we confirm that the species nomenclature throughout the manuscript strictly follows the most recent taxonomic guidelines cited in the Introduction (reference 2, Akhondi et al., 2016), and we have carefully verified consistency across the text.

We thank the reviewer again for these constructive comments, which have strengthened both the current manuscript and the future direction of this project.

Final Comment to Reviewers:

We sincerely thank the reviewers for the thorough and constructive evaluation of our manuscript. The comments provided have been extremely helpful in improving the clarity, methodological transparency, and overall quality of the work. We have carefully addressed all points raised, revising the text, figures, and supplementary materials where appropriate. We believe that these revisions have significantly strengthened the manuscript and enhanced its scientific rigor.

Re: Spectrum03665-25R1 (An integrated molecular approach for the detection and species-level identification of *Leishmania* spp. in clinical samples)

Dear Dr. Paula Ruybal:

Your manuscript has been accepted, and I am forwarding it to the ASM production staff for publication. Your paper will first be checked to make sure all elements meet the technical requirements. ASM staff will contact you if anything needs to be revised before copyediting and production can begin. Otherwise, you will be notified when your proofs are ready to be viewed.

Sincerely,
Ana Cabrera
Editor
Microbiology Spectrum

Reviewer #2 (Comments for the Author):

The authors have addressed the reviewers' concerns. The revised manuscript is now featuring pertinent additions to both the Introduction and Discussion sections, along with enhanced methodological transparency. The authors clarified how the nine hsp70 variants were defined (based on molecular signatures and sequence types) and how species assignment was established via phylogenetic analysis. Directly referencing Supplementary Table 1 in the Results section resolves the previous lack of an explicit link between clinical samples and identification results. Although a sequence alignment was not included in the main text (with the authors arguing the marker's extensive validation), the phylogenetic tree (Figure 4) and supplementary data are sufficient for independent verification.

The authors now explicitly state that the target is the hsp70 type I gene (lines 113 and 246). The explanation for choosing this locus, particularly regarding the absence of type II in *L. braziliensis*, demonstrates scientific rigor and fully satisfies the request. The inclusion of seminal works by Figueira et al. and Requena et al. has strengthened the theoretical framework for using hsp70 as a marker.

They have adopted a more cautious tone when discussing sensitivity and specificity (calculated relative to microscopy, an imperfect gold standard). The addition of a dedicated limitations paragraph (single-center study, retrospective nature, low N for Visceral Leishmaniasis) in lines 431-440 is a positive step that enhances the study's credibility.

Mentioning the potential use of Latent Class Models for future analyses (lines 438-440) shows that the authors took advanced methodological suggestions into account, even if they could not apply them retroactively to the current dataset.